# E-Professionalism among Dental Students from Malaysia and Finland

**DOI:** 10.3390/ijerph19063234

**Published:** 2022-03-09

**Authors:** Pentti Nieminen, Eswara Uma, Shani Ann Mani, Jacob John, Marja-Liisa Laitala, Olli-Pekka Lappalainen

**Affiliations:** 1Medical Informatics and Data Analysis Research Group, University of Oulu, 90014 Oulu, Finland; 2Faculty of Dentistry, Manipal University College Malaysia, Melaka 75150, Malaysia; eswara.uma@manipal.edu.my; 3Department of Paediatric Dentistry and Orthodontics, Faculty of Dentistry, Universiti Malaya, Kuala Lumpur 50603, Malaysia; shani@um.edu.my; 4Department of Restorative Dentistry, Faculty of Dentistry, Universiti Malaya, Kuala Lumpur 50603, Malaysia; drjacob@um.edu.my; 5Research Unit of Oral Health Sciences, Faculty of Medicine, University of Oulu, 90014 Oulu, Finland; marja-liisa.laitala@oulu.fi; 6Medical Research Center, Oulu University Hospital, 90014 Oulu, Finland; 7Department of Oral and Maxillofacial Diseases, Faculty of Medicine, University of Helsinki, 00014 Helsinki, Finland; olli-pekka.lappalainen@helsinki.fi; 8Helsinki University Hospital, 00014 Helsinki, Finland

**Keywords:** dentistry, dental education, dental students, professionalism, social media, Malaysia, Finland

## Abstract

The increased use of social media in dentistry is associated with both advantages and disadvantages. A new form of professionalism, “e-professionalism,” has emerged. It includes an online persona and online information in any format that displays cues to professional identity, attitudes, and behaviors. The objective was to explore the perceptions of Malaysian and Finnish dental students on e-professionalism. A survey of 613 Malaysian and Finnish students was performed. The main variables assessed were posting of objectionable or inappropriate content among students, attitudes towards unprofessional online content, perceived online presence, contacts with patients and faculty members on social media, and concerns about social media use. The prevalence of posting clearly unprofessional content was not high among dental students. Revealing information of patients was most common content of clear unprofessionalism. Students from Malaysia contacted patients and faculty members more actively in social media than students in Finland (73.6% of students in Malaysia and 11.8% in Finland had invited faculty members to be “friends”). Malaysian students were more concerned and more likely to react to inappropriate content on social media. Attitude of dental students towards social media use in dentistry were very positive in both countries. Students agreed that guiding patients online is a new responsibility for dentists in the digital age (86.4% of students in Malaysia and 73.4% in Finland). The findings indicate the existence of both benefits and dangers of social media on e-professionalism among students. There is a need to include robust digital professionalism awareness training for students.

## 1. Introduction

Traditional professionalism refers to the conduct and behavior of the individual in upholding the social contract between society and their profession [1]. The increasing global use of social media, such as social networking, media sharing, blogging, and tweeting, among health care workers assists professional networking, collaboration, education, and training [2]. Concurrently, the perils include blurred professional boundaries, loosening accountability, and compromising confidentiality, impacting the traditional concepts of medical professionalism, ethics, and privacy [2,3]. Consequently, a new form of professionalism, “e-professionalism,” has emerged. Cain and Romanelli [4] defined e-professionalism as the attitudes and behaviors reflecting traditional professionalism paradigms but manifested through digital media.

A number of studies have assessed e-professionalism among various health care workers [2,5]. Unprofessional content on social network by health care workers can range from revealing information about patients, disrespectful statements of colleagues, to profanity, and may be perceived differently based on cultural background and context [6]. Uncertainties among dental students perceived unprofessionalism on online platforms could be attributed to the phenomenon of ‘context collapse’, where online communications feel like an alternative space to daily life, where they can say things they would not say in the real world [7].

Prospective patients are more likely to search for dental professionals through social media networks now than ever before. Many dental practice businesses’ social media accounts are resourceful, giving details of the clinic working hours, scope of service provided and, in some cases, connect with patients using live chats. In an effort to regulate e-professionalism, organizations have published guidelines for professionals [8,9]. In 2020, according to the international guidelines, the Finnish Medical Association published updated ethical rules for social media usage for medical and dental doctors in Finland [10]. Similarly, the Ministry of Health of Malaysia published guidelines use of social media for health care providers in 2016 [11]. In Finland, the national Young Dentist study is carried out every fourth year to explore young professionals’ skills and practices. Unfortunately, the questionnaires do not still include questions related to e-professionalism [12]. The emphasis on inculcating and assessing professionalism in the dental curriculum has increased over the years, which is undoubtedly a complex and context-driven phenomenon [13].

Research into preferred approaches to digital professionalism training prior to graduation is ongoing [14]. Recently, a scale developed to measure e-professionalism among medical and dental students in Croatia had seven factors assessing the following aspects; ethical aspects, dangers of social media, excluding physicians (attitude to towards prohibiting or restricting usage of social networks to medical workers), freedom of choice (independence of acting and posting from university rules and social rules), importance of professionalism, physicians in the digital age, and negative consequences [15]. Unprofessionalism was found to be higher in clinical years among dental students in Thailand [6], while in the UK, there was contradictory perceptions of what constitutes as professional in the online context [7]. A quarter of Greek dental students were friends with their patients on Facebook in their clinical years, of which 58% discussed topics related to dentistry [16]. Hence, the concept of e-professionalism varies widely between different parts of the world.

Differences in social trends, cultural beliefs, and perceptions between Asian and European countries can affect the use of different social media applications. We have recently reported findings about social media usage among dental students in Malaysia and Finland [17]. We found that there were country-specific differences among students in the familiarity and usage of the social media platforms. Extensive use of social media can also be a distraction, and include unprofessional behaviour in the media. Thus, we carried out a descriptive questionnaire study among Malaysian and Finnish dental students to assess perceptions of e-professionalism, which included the online posting of unprofessional content, and actions taken when unprofessional content was viewed.

The research questions were as follows:How widespread is the posting of objectionable or inappropriate content among students?What action do students take when they observe unprofessional online content?How students perceived their online presence?How actively students contact patients and faculty members on social media?What concerns dental students have about social media use?What is the attitude of dental students towards social media use in dentistry?

The main aim of our study was to compare the perceptions of Malaysian and Finnish dental students on e-professionalism. Consequently, we hypothesized that the practices and perceptions of e-professionalism will vary widely between dental students of Malaysia and Finland.

## 2. Materials and Methods

### 2.1. Participants

The study participants comprised of dental under-graduate students from two dental schools in Malaysia (Manipal University College Malaysia and University of Malaya) and Finland (University of Helsinki and University of Oulu) each. In both countries, the dental curriculum is five years. In our previous study, we compared the usage of different social media platforms among these dental undergraduate students [17]. We found that the same top five platforms (WhatsApp, YouTube, Instagram, Facebook, and Snapchat) were the most familiar services to the respondents from both countries. The study sample included students from all five years of study. All undergraduates in the dental schools were invited by email and WhatsApp to participate in this study. Students were assured that participation in the study was voluntary, and protection of confidentiality was guaranteed. Students did not receive benefits or credits for participating in the study.

The study was approved by the Medical Ethics Committee, Faculty of Dentistry, University of Malaya [DF CD2105/0015 (L)] and Research Ethics committee, Faculty of Dentistry, Melaka Manipal Medical college [MMMC/FOD/AR/E C-2021(F-01)] prior to commencement of the study. In Finland, the Ministry of Education and Culture has instructed that those surveys conducted with anonymous questionnaires do not need to be approved by an Ethics Committee.

### 2.2. Questionnaire

All participants were asked to complete a self-administered questionnaire. The instrument used was a questionnaire modified from a previous study among dental students in the USA [5]. This instrument was based on the existing literature and input from focus groups. In addition, the survey was piloted. The questionnaire included three parts. Part 1 consisted of variables related to the basic background factors of the participants. Part 2 included questions related to the social media usage in general, and Part 3 included questions related to e-professionalism. We have reported findings about social media use in a previous study [17]. In this article, we focus on perceptions of e-professionalism and on the challenges that dental students may face with the use of communication technologies in dental education. The questions and variables used in this study are described in the Appendix A.

In Malaysia, an English version of the questionnaire was used. The questionnaire was translated into Finnish. Details about the pre-testing and estimated sample size are reported in our previous article [17]. 

The final questionnaire was administered using Google Forms from 23 March 2021 to 11 April 2021, in both countries, with the link being circulated via email and WhatsApp to student representatives of each year of study, who then forwarded it to their classmates. In the online Google form, all the participants were asked to declare that they had read the participant information sheet (PIS) and voluntarily give consent for data collection and processing. If they refused consent, the questionnaire was closed. Inclusion and exclusion criteria were specified in the PIS, which is included as the Appendix A. The survey was anonymous and did not include sensitive personal data.

### 2.3. Data Analysis

Tabular and graphical displays of data were used as the main tools of data presentation and analysis. The frequency and percentage distributions of participant characteristics (age, sex, year of study, monitoring own online presence, and checking social media for own photos), and all outcome variables were presented for students by country. A chi-square test with exact *p*-values was used to evaluate the statistical significance of differences between Malaysian and Finnish student groups in the frequency tables. Our data conformed to the assumptions and preconditions of the applied analysis method. A *p*-value < 0.05 was regarded as statistically significant. All data analyses were performed using IBM SPSS Statistics software (version 26) (IBM Corp. Armonk, NY, USA) and Origin 2020 graphing software (OriginLab, Northampton, MA, USA).

## 3. Results

### 3.1. Participants

Table 1 shows the distribution of basic characteristics of the 613 participants by country. The Malaysian and Finnish students differed considerably in terms of age and years of dental study. The Malaysian dental students were significantly younger than the Finnish students. Malaysian participants monitored their own online presence more frequently, and checked social media for their own photos. More than 50% of Finnish students did not monitor their visibility on social media (Table 1).

### 3.2. Unprofessional Content

Figure 1 presents the prevalence of self-reported online postings and postings by classmates. Most of the students reported that they had never posted inappropriate material. Among students from Finland, posting objectionable or unsuitable content was statistically significantly more common, especially posting of obscene language, depictions of intoxication, and suspicious material. Only the posting of unidentifiable patient information and discriminatory language was more prevalent among Malaysian students (Figure 1a). Students from both countries reported that they had seen their fellow students posting inappropriate material more often than they had done themselves (Figure 1b). The Malaysian students were less likely to report seeing inappropriate material published by their classmates (*p* < 0.001). A noteworthy detail is that the Finnish students had often noticed that their classmates had posted material showing intoxicated people (44% vs. 11%).

Table 2 shows the distribution of measures taken by the students if they had found information that they believe should not be publicly available. Malaysian students reacted more strongly to such a situation compared to Finnish dental students. In generally, students reported that they had stopped “following” people (47.3% of all respondents) or removed their own names from photos that were tagged to identify them (54.0%).

Most of the students in both countries had not noticed that their online presence was inaccurate, incomplete, unprofessional, or absent (Figure 2). However, a higher percentage of Malaysian students than Finnish students reported to have noticed at least one of these issues in their online presence (41% vs. 24%, *p* < 0.001). In Malaysia, 18% of students had found their online presence as unprofessional, but in Finland, this was only 9% (*p* = 0.002).

### 3.3. Social Media Invitations

Table 3 shows the proportions of social media invitation activity by country. A higher percentage of Malaysian students had accepted invitation(s) from patients to be “friends”, compared to Finnish students (14.1% vs. 1.0%, *p*-value of chi-square test <0.001). A total of 6.1% of students from Malaysia reported that they have invited patients to be “friends”, while only one student from Finland (0.5%) reported such activity. Among Malaysian students, the online activity also included accepting invitations from faculty members to be “friends” (84.3%) and inviting faculty members to be “friends” (73.6%). In the Finnish universities, this activity was much less frequent: only 11.8% of the Finnish students reported that they had invited faculty members to be “friends”. Googling faculty members was common in both countries. A higher percentage of Finnish students had Googled faculty members compared to Malaysian students (80.8% vs. 63.4%, *p*-value of chi-square test <0.001).

### 3.4. Concerns about Social Media Use

Table 4 shows that Malaysian and Finnish dental students were statistically significantly differently concerned about the use of social media. Most Malaysian dental students (95.7%) ranked public perceptions of unprofessional behavior by them as a somewhat or very important concern about social media use, compared with 40.4% of Finnish students. In addition, 94.5% of Malaysian dental students ranked public perceptions of unprofessional behavior by classmates as a somewhat or very important concern about social media use, compared with 49.7% of Finnish students. Almost all Malaysian students (95.9%) rated violations of patient confidentiality as somewhat or very important concerns about social media use. Finnish students did not see violations of confidentiality as such a big threat, and 46.8% of them were not concerned at all. In both countries, students were concerned about publishing incorrect dental information to patients. Here, too, the Malaysians rated the risk of misinformation in social media to be higher than the Finns did.

### 3.5. Benefits of Social Media on E-Professionalism of Health Care Professionals

Almost all Malaysian dental students (90.7%) and the majority of Finnish students (83.7%) agreed or strongly agreed that patients use social media to obtain dental information (*p* = 0.038) (Figure 3). Most Malaysian respondents (82.3%) agreed or strongly agreed that the benefits of social media use in dentistry outweigh its risks, compared with 69.5% of Finnish respondents (*p* < 0.001). Similar differences were observed between Malaysian and Finnish dental students in their responses to the statements “As a student of dentistry, it is my obligation to keep up to date on social media use” (93.4% vs. 68.9%, respectively) or “Guiding patients online is a new responsibility for dentists in the digital age” (86.4% vs. 73.4%, respectively) (Figure 3).

## 4. Discussion

E-professionalism is defined as the attitudes and behaviors reflecting traditional professionalism paradigms, but manifested through digital media [4]. The present descriptive paper investigated how Malaysian and Finnish dental students reacted to unprofessional online content, how concerned they were about unprofessional behavior in the media, and how they perceived the potential benefits of social media use in dentistry. We found that most of the students from both countries reported that they had never posted inappropriate material. Students from Malaysia more actively contacted patients and faculty members in social media than students from Finland. Malaysian students were more concerned about the use of social media, and were more likely to react to inappropriate content. In general, the attitudes of dental students towards social media use in dentistry were very positive in both countries. We also found that students agree that digital technology is impacting on the training and development of dental professionals.

Numerous studies have tried to assess e-professionalism by health care professionals, students, and their colleagues [2]. Although there is no uniform consensus on what constitutes unprofessional behavior, studies most frequently associate it with online content pertaining to information about patients, obscene language, alcohol intoxication, substance or illegal drug use, sexually suggestive material, discrimination, or bullying of classmates. Surveys that captured self-reported online behavior among students from different disciplines have reported varying frequencies of unprofessional contents [2]. Our study is in line with previous studies that the prevalence of clearly unprofessional content is not high among dental students [6,16,18].

Asian society is heavily influenced by a handful of Asian religions and philosophies such as Buddhism, Daoism, Confucianism, Hinduism, and Islam, among others, which have influenced the Asians ontologically and epistemologically [19]. In European Nordic countries, many people are still nominally a member of a church; however, for most of them, this is more about an act of citizenship than it is about religious belief. Religion is largely treated as a private matter, and most are of the opinion that you should be free to believe whatever you want [20]. Another essential difference between Western and Asian society is the position of the individual [21]. While Western culture is characterized by a strong individualistic self-image, in the Asian context, group consciousness plays a much bigger part. They obtain their identity from the position they hold in the group. It is characterized as subjugation of individual interest and inclination to strong hierarchical authority [19]. From the perspective of European Nordic countries, freedom for the individual is always the goal of human communication. Individuals would mobilize all necessary human or natural resources towards the freedom of self and others. In summary, Asian societies stress the interests of the community, and Western societies stress the importance of the individual [19,21].

In Malaysia, posting of sexually suggestive material or depiction of intoxication was reported to happen only occasionally. On the other hand, in Finland, posting of excessive alcohol consumption was common among dental students. Karveleas et al. [16] investigated recently the relationship between Facebook behavior and e-professionalism. In their study, posting holiday pictures (72%) or wearing swimwear (26%) was categorized as unprofessional. In Finland, for example, this would not be interpreted as unprofessional content. This illustrates the complexity of the concept of unprofessionalism and its possible dependence on cultural ties.

Students from Malaysia more actively contacted patients through social media than students in Finland. This disproportion in range could be explained by cultural and regulatory differences. One reason for the difference might be the strict legislation on the EU area concerning the individuals’ data protection. Although online interaction (“friending”) with a patient is generally not acceptable nor endorsed, a wide range of opinions have been reported concerning this issue [22,23].

In Malaysia it was common to contact and invite faculty members to “friends” in social media. Thus, social media is used for work-related matters as well as personal connections. Moreover, eventually all students will inevitably join the dental fraternity, and social media will be used as a tool to stay in touch. Our findings showed that willingness to share your private life outside of the study work was low in Finland. In addition, both students and faculty could be worried that connecting via social media would increase the conflict between personal and private lives, and mix the boundaries of the teacher-student relationship [24,25].

Social media poses new dangers for many dental patients, especially younger people [26,27,28]. Social media provides unsupervised access to images of current trends in beauty and fashion. There are commercially driven advertisements for dubious treatments promoted or dental quackery on the social media which are not based on sound scientific evidence [27,28]. Some dental influencers are endorsing questionable products or advertisements for payments without making relevant conflict of interest declarations. Advertising ‘perfect smiles’ or ‘ideal’ facial images on social media may be adversely affecting vulnerable viewers [26]. Social media usage in dentistry is also thought to be contributing towards increased comparisons, envy, low self-esteem, body dysmorphia, and social anxiety [29]. Rana and Kelleher [28] have focused on younger dental patients. They note that social media images young patients are exposed to have some benefits, but others can pose significant risks for them (i.e., self-esteem issues, unrealistic expectations, growing up to think that a ‘Hollywood smile’ or face is the ideal, bullying or peer pressure). Most of the participants in our study had understood that guiding the younger dental patients online is important.

Our findings showed that students inadequately acknowledged the potential threats of inaccurate dental information in social media for patients. On an educational level of e-professionalism, dental schools have the responsibility to teach their dental students and dentists various knowledge management methods to cope with online scientific information. Dental students should be more knowledgeable than their patients are, and know the sources of inaccurate information [30]. Majority of the participating students from both countries agreed that guiding patients online is a new responsibility for dentists.

In our questionnaire, we did not ask students’ attitudes towards possible regulation of their online activity by bodies, such as the dental school or dental associations. Gormly et al. [14] found in their qualitative study that dental students in the UK expressed a declaration of autonomy with respect to their online activity. This autonomy was expressed as a rejection of professional regulation, and claimed that they are best placed to evaluate their online activities. It is common and understandable for medical and dental undergraduates to push back against regulation and defend the need for free speech. Our research showed that students assessed their online visibility, and changed their profile where necessary. Most of them did not find their online presence to be unprofessional, or would have made changes where necessary.

The present cross-sectional study used a self-completed online questionnaire and may suffer from issues related to the validity and reliability of the instrument. Following the completion of the present study, a recent research paper including a validated SMePROF-S scale questionnaire by Marelic et al. [15] was published. Our survey included questions touching similar topics: ethical aspects, dangers of social media, freedom of choice in social media and the importance of social media on professionalism, and physicians of the digital age, as well as negative consequences of social media on profession. We also note that, from an international perspective, our study was performed in national contexts in Malaysia and Finland, and had a limited study population, although representing more than a half of the Finnish dental student population and covering geographically all parts of the country. We believe that the findings could be generalizable beyond the immediate contexts.

There is a need for dental school professionalism curricula to include robust digital professionalism awareness training for students. On an educational level for students, recommendations should include a variety of e-professionalism topics into a curriculum to provide students with a clear picture of what constitutes professional violations on social media, and assist them in distinguishing between personal and professional personas online [2,14,16,31,32]. Exploring the digital relationships between teachers and students would be an interesting area for future research. Research on the attitudes and practices of the social media behavior among the university teachers in social media is also important [32].

## 5. Conclusions

The findings indicate the existence of both benefits and dangers of social media on e-professionalism among students. The practices and concerns regarding e-professionalism seemed to differ remarkably between the two countries. It is always important to consider the influence of culture on individuals’ attitudes and behavior when determining the reactions to sensitive material on social media. Our study clearly indicates the need for increased social media regulation at any stage of the dental education.

## Figures and Tables

**Figure 1 ijerph-19-03234-f001:**
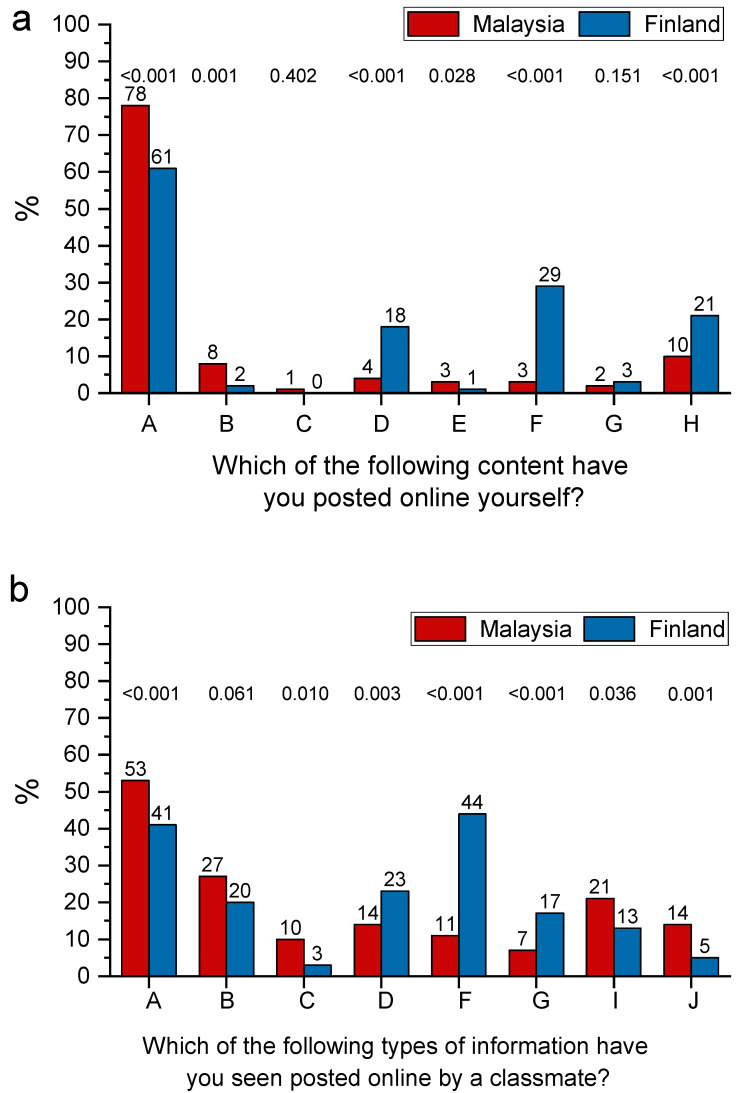
Percentage distributions of responses to the questions “Which of the following content have you posted online yourself?” (**a**) and “Which of the following types of information have you seen posted online by a classmate?” (**b**) by country. The contents are as follows: A = None, B = Unidentifiable patient information, C = Identifiable patient information, D = Obscene language, E = Discriminatory language, F = Depiction of intoxication, G = Sexually suggestive material, H = Items I thought were initially appropriate, but for various reasons later took down, I = Items I found objectionable but did not discuss with my classmate, J = Items I found objectionable and did discuss with my classmate. Data include dental undergraduate students from Malaysia (n = 440) and Finland (n = 203). Statistical significances between countries were evaluated by chi-square test.

**Figure 2 ijerph-19-03234-f002:**
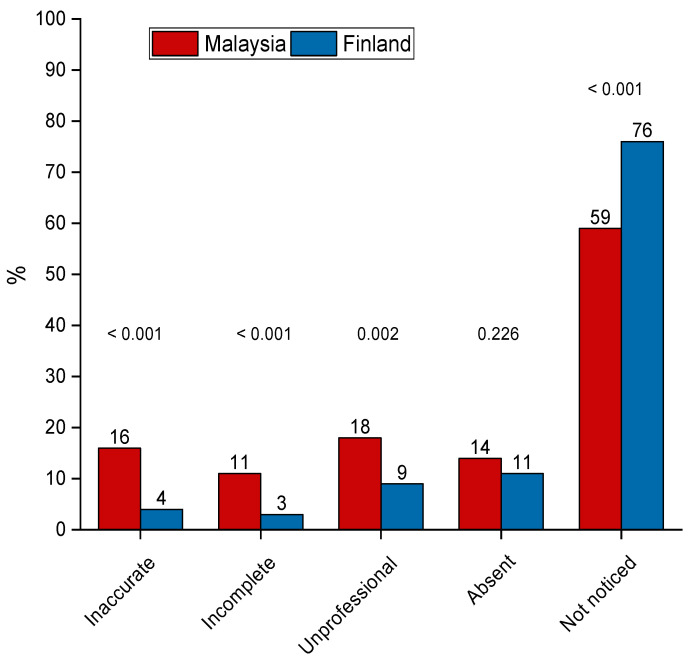
Percentage distributions of responses to the question “Have you ever found that your online presence was any of the following?” Data include dental undergraduate students from Malaysia (*n* = 440) and Finland (*n* = 203). Statistical significances between countries were evaluated by chi-square test.

**Figure 3 ijerph-19-03234-f003:**
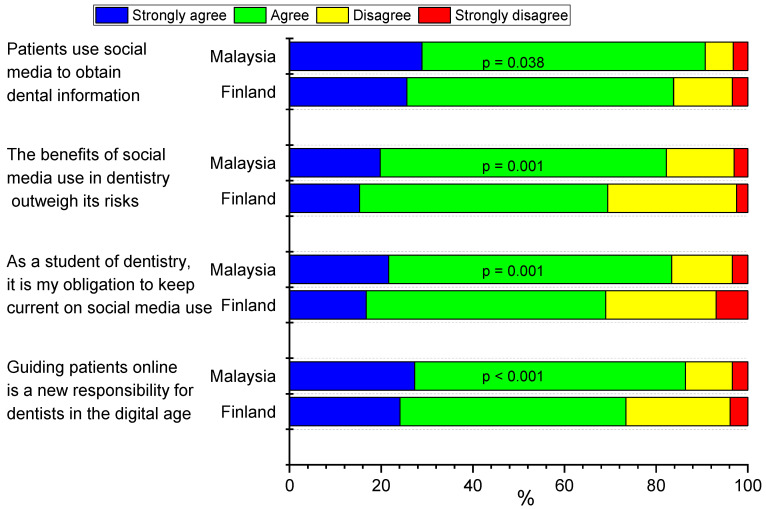
Percentage distributions of responses to the question “To what extent do you agree or disagree with the following statement?” by country. Data include dental undergraduate students from Malaysia (*n* = 440) and Finland (*n* = 203). Statistical significances between the countries were evaluated by chi-square test.

**Table 1 ijerph-19-03234-t001:** The frequency and percentage distributions of basic characteristics among study participants from Malaysia (*n* = 440) and Finland (*n* = 203). Statistical significances between countries were evaluated by chi-square test.

	Country			*p*-Value of Chi-Square Test
	Malaysia	Finland	All	
Characteristics	*n* (%)	*n* (%)	*n* (%)	
Age				<0.001
▪ 20 years or younger	70 (17.0)	12 (5.9)	87 (13.5)	
▪ 21–23 years	250 (56.8)	56 (27.6)	306 (47.6)	
▪ 24–26 years	114 (25.9)	69 (34.0)	183 (28.5)	
▪ 27–29 years	1 (0.2)	38 (18.7)	39 (6.1)	
▪ 30 years or above	0	28 (13.8)	28 (4.4)	
Sex				>0.999
▪ Male	103 (23.4)	47 (23.2)	150 (23.3)	
▪ Female	337 (76.6)	156 (76.7)	493 (76.7)	
Year of dental school				0.038
▪ First	70 (15.9)	39 (19.2)	109 (17.0)	
▪ Second	78 (17.7)	54 (26.6)	132 (20.5)	
▪ Third	79 (18.0)	33 (16.3)	112 (17.4)	
▪ Fourth	108 (24.5)	42 (20.7)	150 (23.3)	
▪ Fifth	105 (23.9)	35 (17.2)	140 (21.8)	
Monitoring own online presence				<0.001
▪ Never	126 (28.6)	107 (52.7)	233 (36.2)	
▪ Occasionally	187 (42.5)	96 (47.3)	283 (44.0)	
▪ Regularly	75 (17.0)	0	75 (11.7)	
▪ Frequently or very frequently	52 (11.9)	0	52 (8.1)	
Checking social media for own photos				<0.001
▪ Never	82 (18.6)	118 (58.1)	200 (31.1)	
▪ Occasionally	203 (46.1)	83 (40.9)	286 (44.5)	
▪ Regularly	119 (27.0)	2 (1.0)	121 (18.8)	
▪ Frequently or very frequently	36 (8.1)	0	36 (5.6)	

**Table 2 ijerph-19-03234-t002:** Frequency and percentage distributions of responses to the question “What action have you taken if you have found information that you believe should not be publicly available?” by country. Data include dental undergraduate students from Malaysia (*n* = 440) and Finland (*n* = 203).

	Country			
	Malaysia	Finland	All Students	*p*-Value of Chi-Square Test
Variable	*n* (%)	*n* (%)		
Deleted people from my “friends” list				<0.001
▪ No	243 (55.2)	143 (70.4)	386 (60.0)	
▪ Yes	197 (44.8)	60 (29.6)	257 (40.0)	
Stopped “following” people				<0.001
▪ No	202 (45.9)	137 (67.5)	339 (52.7)	
▪ Yes	238 (54.1)	66 (32.5)	304 (47.3)	
Deleted comments made by others on my profile				<0.001
▪ No	260 (59.1)	153 (75.4)	413 (64.2)	
▪ Yes	180 (40.9)	50 (24.6)	230 (35.8)	
Removed my name from photos that were tagged to identify me				0.075
▪ No	192 (43.6)	104 (51.2)	296 (46.0)	
▪ Yes	248 (56.4)	99 (48.8)	347 (54.0)	

**Table 3 ijerph-19-03234-t003:** Frequency and percentage distributions of social media activities by country. Data include dental undergraduate students from Malaysia (*n* = 440) and Finland (*n* = 203).

	Country			*p*-Value of Chi-Square Test
	Malaysia	Finland	All Students	
Variable	*n* (%)	*n* (%)		
I use privacy settings				0.852
▪ No	24 (5.5)	10 (4.9)	34 (5.3)	
▪ Yes	416 (94.5)	193 (95.1)	609 (94.7)	
I have accepted invitation(s) from patients to be “friends”				<0.001
▪ No	378 (85.9)	201 (99.0)	579 (90.0)	
▪ Yes	62 (14.1)	2 (1.0)	64 (10.0)	
I have invited patients to be “friends”				<0.001
▪ No	413 (93.9)	202 (99.5)	615 (95.6)	
▪ Yes	27 (6.1)	1 (0.5)	28 (4.4)	
I have accepted invitations by faculty members to be “friends”				<0.001
▪ No	67 (15.2)	120 (59.1)	187 (29.1)	
▪ Yes	373 (84.8)	83 (40.9)	456 (70.9)	
I have invited faculty members to be “friends”				<0.001
▪ No	116 (26.4)	179 (88.2)	295 (45.9)	
▪ Yes	324 (73.6)	24 (11.8)	348 (54.1)	
I have googled faculty members				<0.001
▪ No	161 (36.6)	39 (19.2)	200 (31.1)	
▪ Yes	279 (63.4)	164 (80.8)	443 (68.9	

**Table 4 ijerph-19-03234-t004:** Frequency and percentage distributions of responses to the question “What concerns do you have about social media use?” by country. Data include dental undergraduate students from Malaysia (*n* = 440) and Finland (*n* = 203).

What Concerns Do You Have about Social Media Use?	Country	Not at All Important *n* (%)	Somewhat Important *n* (%)	Very Important *n* (%)	*p*-Value of Chi-Square Test
Public perceptions of unprofessional behavior by me	Malaysia Finland	19 (4.3) 121 (59.6)	188 (42.7) 71 (35.0)	233 (53.0) 11 (5.4)	<0.001
Family perceptions of unprofessional behavior by me	Malaysia Finland	52 (11.8) 176 (86.7)	208 (47.3) 23 (11.3)	180 (40.9) 4 (2.0)	<0.001
Public perceptions of unprofessional behavior by my colleagues	Malaysia Finland	24 (5.5) 102 (50.2)	220 (50.0) 94 (46.3)	196 (44.5) 7 (3.4)	<0.001
Public perceptions of my dental school	Malaysia Finland	21 (4.8) 150 (73.9)	207 (47.0) 44 (21.7)	212 (48.2) 9 (4.4)	<0.001
Public perceptions of the dental profession	Malaysia Finland	19 (4.3) 106 (52.2)	185 (42.0) 85 (41.9)	236 (53.6) 12 (5.9)	<0.001
Violations of patient confidentiality	Malaysia Finland	18 (4.1) 95 (46.8)	149 (33.9) 82 (40.4)	273 (62.0) 26 (12.8)	<0.001
Posting of inaccurate dental related information for patients	Malaysia Finland	7 (1.6) 67 (33.0)	130 (29.5) 96 (47.3)	303 (68.9) 40 (19.7)	<0.001

## Data Availability

The data presented in this study are available on request from the corresponding author.

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
