# Peer review of "E-Professionalism among Dental Students from Malaysia and Finland"

_ijerph, 2022, doi:10.3390/ijerph19063234_

Round 1
Reviewer 1 Report
Originality/Novelty:
- The study is a unique, novel one in that it focuses on Malaysia and Finland dental students. Well done to the authors for their efforts in conducting this study. The authors present an example that is likely to be of relevance to the international readership. Nevertheless, the authors must provide reasons why the study concentrates specifically on Malaysia and Finland dental students. This should be explained within lines 82-86.
Significance of content:
- An interesting point has been made on lines 282-283: “This illustrates the complexity of the concept of unprofessionalism and its possible dependence on cultural ties.” Please expand on the concept of “cultural ties.” Are there any particular cultural or societal ties, customs or traditions that might explain the differences in the results obtained from the Malaysia and Finland dental students? If so, please consider these when discussing the results. The paper shall be more strengthened, if the authors can consider the reasons behind the results.
- The paper appears more supportive and positive towards social media usage in dentistry. However, the authors need to create more of a balance, and there needs to be more discussion of the pitfalls. After the sentence: “Majority of the participating students from both countries agreed that guiding patients online is a new responsibility for dentists” (lines 304-305), there needs to be a discussion on why guiding patients online is a new responsibility for dentists. A recent, relevant publication that can be referenced:- Modha, B., 2021. Collaborative leadership with a focus on stakeholder identification and engagement and ethical leadership: a dental perspective. British Dental Journal, 231(6), pp.355-359, highlights how within social media, there are self-aggrandising advertisements and claims for egotistical, narcissistic dentistry, often promoted by younger, or apparently non-specialist dentists; how some dental influencers are endorsing questionable products or advertisements for payments without making relevant conflict of interest declarations; how social media usage in dentistry is thought to be contributing towards increased comparisons, envy, low self-esteem, body dysmorphia and social anxiety; how self-descriptive designations such as 'cosmetic dentist' used by many dentists on social media without specialist registration is misleading patients, and how advertising ‘perfect smiles’ or ‘ideal’ facial images on social media may be adversely affecting vulnerable viewers that could have a facial disfigurement or body dysmorphia condition relating to their teeth, mouth or face. These important points made by Modha (2021) would very much support the statement made on lines 304-305. Please also consider how these points can affect dental students who are studying at dental school. Another very relevant publication that can be referenced:- Rana, S. and Kelleher, M., 2018. The Dangers of Social Media and Young Dental Patients' Body Image. Dental Update, 45(10), pp.902-910 – this article focuses on the younger generation, and the authors should consider how younger dental patients may be at risk (i.e., self-consciousness, self-esteem issues, unrealistic expectations, growing up to think that a ‘Hollywood smile’ or face is the ideal, bullying, peer pressure, etc), and how guiding these younger dental patients online shall be important. This is of relevance to dental students also, so please consider how young dental students may also be affected by this.
Quality of presentation:
- Overall, the paper is well-written and fairly easy to read. It is well structured and laid out. The figures, tables and statistics are satisfactory. The utilised references are fine. However, please kindly note, the language, grammar, punctuation, spelling and sentence structures within the current paper, all must be thoroughly assessed and polished to ensure a succinct and coherent read. This is because there are minor discrepancies in the language, grammar, punctuation, spelling and sentence structures. Please obtain the necessary scientific English language reading and editing assistance, if need be, so that the paper has the potential to be read enjoyably by the international readership.
Scientific soundness:
- There are no obvious issues pertaining to scientific soundness. This article appears to have a satisfactory scientific basis.
- Please consider adding the following keyword: dental students.
Interest to readers:
- It would be expected that this paper shall be of interest to other dentists and allied dental professionals working within the dental education sector. This article may be of interest to healthcare professionals involved within teaching and mentoring lines of work.
Overall merit:
- Thank you for the opportunity to review this manuscript. It was interesting and thought provoking to read. It does have merit. However, if the aforementioned additions and changes are made, then this paper shall have much more potential for publication.
Reviewer 2 Report
The manuscript is well written and presented. Is is not clear why Finland and Malaysia were compared! This is not elaborated upon in the introduction nor in the discussion . This is also important since cultural background is also different. I understand the nature of multicenter studies but in this case, this needs to be adequately approached and made clear. The message will be much easier if 2 seperate studies one in each country.
The research questions listed at the end of the introduction section can be presented in a paragraph with appropriate citations and moved before the hypothesis paragraph
Reviewer 3 Report
Dear authors,
thank you for the opportunity to review this manuscript treating a novel and relevant topic. However, there are by now several major points that make the manuscript not suitable for publication.
These points and also some minor formal points are listed below:
General:
It is not clear, why you chose the country combination Malaysia and Finland. Furthermore, a validated scale or score should be used, if you plan to create data suitable for statistical analysis. At this stage, the manuscript is very descriptive and does not work with a clear hypothesis.
Introduction:
L. 64: Doctors and dentists should, from an international point of view, not be distinguished linguistically, as these terms overlap. Please be more concise.
L. 74: What does "excluding physicians" and "freedom of choice" mean in this context (maybe provide a bracket with an explanation, as this points seems odd to the reader)
L. 80 and 72: As you wrote "Hence, the concept of e-professionalism varies widely between different parts of the world." You should provide an information about the origin of the described e-professionalism scale in Line 72.
L. 87 ff.: In addition to the research questions, you should add favourably a hypothesis or at least a clear aim of the study at the end of the introduction part.
Mat Met:
At the beginning of your M&M section, you should add information about the definition of social media (which platforms have been included for which country)
L.99: Please close the bracket (typo).
L. 127-132: Please move the ethics statement to the very Beginning of the M&M section (Participants)
L. 133: For the statistical method, please provide the alpha (level of significance) of the statistical tests. In the results part, you should describe values above the alpha only as "not-significant"
L. 139: What does the sentence "We used software which reported exact p values for the chi-square test statistic [18]." to knowledge? Rather remove it.
Results:
In general, the results part is quite descriptive and should provide a clearer answer to the question, how high the e-professionalism of the dental students in the two countries were.
Table 1: It is completely unclear, to which comparison the p-value on the right side belongs. Is it the comparison between Malaysia and FInland? Please add the information in the figure legend.
L. 164: Why is a very specific point as "inoxicated patients" a single criterion? You have to discuss this in the discussion part. Do you mean alcohol intoxation of patients or students? Why is it separated from other inappropriate context?
Fig. 1: Which test belongs to the p-values? What has been compared (probably finnland vs. malaysia again). Please add this information to the figure legend.
L. 213: Are "faculty members" teaching doctors? Please use a more specific term.
L. 216: to google is not an appropriate verb. Please use scientific language (e.g. to research in a search engine)
The statistics in Figure 3 seem to be quite questionable. Finnland and Malaysia answered quite similar. However, almost every Question reveals significant differences. What was tested? Have you testet the "agree" frequency?
L. 284-285: Here, you discuss a very important point. The whole social media behaviour depends on the regulations in the respective country. Thus, you should discuss why you investigated malaysia and Finland and use a general definition of non-professional or inappropriate behaviour .
L. 293-295: You should keep your scientific manuscript free from nationally based prejudices.
L. 322-324: Why do you cover the whole country when you investigate one university?
Round 2
Reviewer 1 Report
Thank you to the authors for their efforts in making additions and changes to enhance the manuscript. I do feel that the manuscript is now at a much better standard to warrant publication. However, although the paper now contains more necessary content, there are minor discrepancies with the language, grammar, punctuation, spelling and sentence structure. This all must be thoroughly assessed and polished throughout to ensure a succinct and coherent read. Please enlist the necessary scientific English language reading and editing assistance if need be, so that the paper has the potential to be read enjoyably by the international readership. Best wishes.
Author Response
Thank you to the authors for their efforts in making additions and changes to enhance the manuscript. I do feel that the manuscript is now at a much better standard to warrant publication. However, although the paper now contains more necessary content, there are minor discrepancies with the language, grammar, punctuation, spelling and sentence structure. This all must be thoroughly assessed and polished throughout to ensure a succinct and coherent read. Please enlist the necessary scientific English language reading and editing assistance if need be, so that the paper has the potential to be read enjoyably by the international readership. Best wishes.
Thank you. The manuscript has now undergone English language editing by the publisher MDPI. The text has been checked for correct use of grammar and common technical terms, and edited to a level suitable for reporting research in a scholarly journal. MDPI uses experienced, native English-speaking editors.
Reviewer 3 Report
Dear authors,
thank you for the renewed possibility to review this interesting manuscript. You took all of my comments on boards.
However, there are some major points that need to be adapted from my point of view:
L. 329: For a sentence like "In Finland willingness to share your private life 329 outside of the study work was low." you definitely need a reference. Please double-check the whole manuscript for sufficient scientific literature references, especially in parts were cultural differences between Malaysia and Finland are discussed.
General: During the review process, the study turned out to be rather descriptive. This point needs to be incorporated into the whole manuscript, including the title, the hypothesis and the discussion.
Author Response
Dear authors, thank you for the renewed possibility to review this interesting manuscript. You took all of my comments on boards.
Thank you.
However, there are some major points that need to be adapted from my point of view:
L. 329: For a sentence like "In Finland willingness to share your private life outside of the study work was low." you definitely need a reference. Please double-check the whole manuscript for sufficient scientific literature references, especially in parts were cultural differences between Malaysia and Finland are discussed.
We have checked the references. The sentence quoted by the reviewer is not a reference to previous research. It refers to our findings in this study. We have now reformulated this sentence as follows:
“Our findings showed that willingness to share your private life outside of the study work was low in Finland.”
General: During the review process, the study turned out to be rather descriptive. This point needs to be incorporated into the whole manuscript, including the title, the hypothesis and the discussion.
We would like to point out that we have used statistical inference methods to evaluate statistical significance. However, we have now emphasized the descriptive nature of our research in the introduction and discussion sections.